# DRUL for school: Opening Pre-K with safe, simple, sensitive saliva testing for SARS-CoV-2

Mayu O. Frank[1], Nathalie E. Blachere[1,2], Salina Parveen[1], Ezgi Hacisuleyman[1], John Fak[1], Joseph M. Luna[1,3], Eleftherios Michailidis[3], Samara Wright[4], Pamela Stark[4], Ann Campbell[5], Ashley Foo[5], Thomas P. Sakmar[6], Virginia Huffman[7], Marissa Bergh[1], Audrey Goldfarb[8], Andres Mansisidor[9], Agata L. Patriotis[10], Karl H. Palmquist[11], Nicolas Poulton[12], Rachel Leicher[13], César D. M. Vargas[14], Irene Duba[9], Arlene Hurley[15], Joseph Colagreco[16], Nicole Pagane[9], Dana E. Orange[1,17], Kevin Mora[1], Jennifer L. Rakeman[18], Randal C. Fowler[18], Helen Fernandes[1,19], Michelle F. Lamendola-Essel[1,20], Nicholas Didkovsky[1], Leopolda Silvera[21], Joseph Masci[21], Machelle Allen[21], Charles M. Rice[3], Robert B. Darnell[1,2]*

1 Laboratory of Molecular Neuro-oncology, The Rockefeller University, New York, NY, United States of America, 2 Howard Hughes Medical Institute, The Rockefeller University, New York, NY, United States of America, 3 Laboratory of Virology and Infectious Disease, The Rockefeller University, New York, NY, United States of America, 4 Child and Family Center, The Rockefeller University, New York, NY, United States of America, 5 Occupational Health Services, The Rockefeller University, New York, NY, United States of America, 6 Laboratory of Chemical Biology and Signal Transduction, The Rockefeller University, New York, NY, United States of America, 7 Human Resources, The Rockefeller University, New York, NY, United States of America, 8 Laboratory of Cell Biology and Genetics, The Rockefeller University, New York, NY, United States of America, 9 Laboratory of Genome Architecture and Dynamics, The Rockefeller University, New York, NY, United States of America, 10 Laboratory of Chromatin Biology and Epigenetics, The Rockefeller University, New York, NY, United States of America, 11 Laboratory of Morphogenesis, The Rockefeller University, New York, NY, United States of America, 12 Laboratory of Host-Pathogen Biology, The Rockefeller University, New York, NY, United States of America, 13 Laboratory of Nanoscale Biophysics and Biochemistry, The Rockefeller University, New York, NY, United States of America, 14 Laboratory of Neurogenetics of Language, The Rockefeller University, New York, NY, United States of America, 15 Rockefeller University Hospital, The Rockefeller University, New York, NY, United States of America, 16 Leonard Wagner Laboratory of Molecular Genetics and Immunology, The Rockefeller University, New York, NY, United States of America, 17 Hospital for Special Surgery, New York, NY, United States of America, 18 Public Health Laboratory, New York City Department of Health and Health and Mental Hygiene, New York, NY, United States of America, 19 Department of Pathology and Cell Biology, Columbia University, New York, NY, United States of America, 20 Memorial Sloan Kettering Cancer Center, New York, NY, United States of America, 21 Elmhurst Hospital Center, New York City Health and Hospitals, Elmhurst, NY, United States of America

* darnelr@rockefeller.edu

**Data Availability Statement:** All relevant data are within the paper and its Supporting Information files.

## Abstract

To address the need for simple, safe, sensitive, and scalable SARS-CoV-2 tests, we validated and implemented a PCR test that uses a saliva collection kit use at home. Individuals self-collected 300 μl saliva in vials containing Darnell Rockefeller University Laboratory (DRUL) buffer and extracted RNA was assayed by RT-PCR (the DRUL saliva assay). The limit of detection was confirmed to be 1 viral copy/μl in 20 of 20 replicate extractions. Viral RNA was stable in DRUL buffer at room temperature up to seven days after sample collection, and safety studies demonstrated that DRUL buffer immediately inactivated virus at concentrations up to $2.75 \times 10^6$ PFU/ml. Results from SARS-CoV-2 positive nasopharyngeal (NP) swab samples collected in viral transport media and assayed with a standard FDA Emergency Use Authorization (EUA) test were highly correlated with samples placed in

**Funding:** C.M.R acknowledges the generous support of the G. Harold and Leila Y. Mathers Charitable Foundation, the BAWD Foundation, and The Rockefeller University. R.B.D. wishes to disclose that he receives consulting fees as a Senior Visiting Fellow at MITRE Corporation, has started a charitable LLC, D4S Testing, to offer free DRUL saliva testing to NYC school children, and is an Investigator of the Howard Hughes Medical Institute.

**Competing interests:** T.P.S. wishes to thank the Richard Lounsbery Foundation, the Elinor Schwartz Charitable Trust and the Danica Foundation for support. DEO is an inventor on a provisional unlicensed patent entitled "Method and System for RNA Isolation from Self-Collected and Small Volume Samples", US 63/050,155. C.M.R acknowledges the generous support of the G. Harold and Leila Y. Mathers Charitable Foundation, the BAWD Foundation, and The Rockefeller University. R.B.D. discloses that he receives consulting fees as a Senior Visiting Fellow at MITRE Corporation, that he has started a charitable LLC, D4S Testing, to offer free DRUL saliva testing to NYC school children, and that he is an Investigator of the Howard Hughes Medical Institute. This does not alter our adherence to PLOS ONE policies on sharing data and materials.

DRUL buffer. Direct comparison of results from 162 individuals tested by FDA EUA oropharyngeal (OP) or NP swabs with co-collected saliva samples identified four otherwise unidentified positive cases in DRUL buffer. Over six months, we collected 3,724 samples from individuals ranging from 3 months to 92 years of age. This included collecting weekly samples over 10 weeks from teachers, children, and parents from a pre-school program, which allowed its safe reopening while at-risk pods were quarantined. In sum, we validated a simple, sensitive, stable, and safe PCR-based test using a self-collected saliva sample as a valuable tool for clinical diagnosis and screening at workplaces and schools.

## Introduction

The SARS-CoV-2 pandemic has raged in the United States, with over 400,000 deaths by the end of Trump administration [1, 2]. Mitigation of this tragedy has struggled alongside the lack of a uniform approach to testing, including mixed messages from the Centers for Disease Control and Prevention (CDC) [3]. These challenges were exacerbated by shortages of testing reagents and supplies [4–6]. Scalable, low cost, accessible testing, in symptomatic and asymptomatic individuals is critical to management of the pandemic. Workplaces and schools need workable strategies to test students, employees and families. Working mothers have been disproportionately affected by the need to care for children who are at home during school closures [7]. Testing is increasingly being used to supplement contact tracing efforts. Collecting, transporting and handling samples in buffer that inactivates virus may decrease exposure risk for healthcare providers and laboratory personnel [8].

Saliva testing is seen as an accessible and scalable means of testing, particularly in the school setting since it does not require technical expertise for collection. However, a wide range of tests have been developed, and those with low sensitivity pose an increased risk of reporting false negatives, which may give a false sense of security and decrease transmission mitigating behaviors. We developed an assay that simplifies sample collection and minimizes contact and exposure, using a kit for self-collection of saliva specimens. The DRUL buffer is based on the solution widely used in RNA extraction that contains guanidine thiocyanate [9]. Samples were assayed using a test developed using the CDC 2019-nCoV Real-Time PCR Diagnostic Panel primers and probes and RT-PCR [10] as authorized by the NY State Clinical Laboratory Evaluation Program (CLEP) for use as a clinical diagnostic test. Here we report the results of our validation and initial implementation of this testing strategy.

## Materials and methods

### Study subjects

Individuals voluntarily participated in sample collection for serial screening. They were provided with a sample collection kit and instructions (S1 Fig). Protocols for the collection of saliva samples were either approved by the Rockefeller University (RU) Institutional Review Board (IRB) and Biomedical Research Alliance of New York IRB or were deemed not to be human subjects research by the RU IRB. Where required, written or verbal consent was obtained from all volunteers.

### Specimen collection and processing

Individuals were instructed to avoid eating or using oral cleansing agents for 30 minutes prior to collection of saliva (or their children's saliva) in a medicine cup, and then transfer 300 μl of

**Table 1. DRUL buffer reagents.**

| Reagent | Amount needed | Purpose | Manufacturer | Catalog # |
|---|---|---|---|---|
| 5M Guanidine Thiocyanate | 59.08 g | Protein denaturing agent, Isolation of RNA | Fisher BioReagents | BP221-1 |
| 0.5% Sarkosyl | 5 ml of 10% Sarkosyl | Cell lysis, detergent | Fisher BioReagents | BP234-500 |
| 25mM Sodium Acetate (3M), pH 5.5 | 0.83 ml | Precipitation of RNA | Invitrogen | AM9740 |
| Nuclease-free Water | Bring up to 100 ml | Dilution | Ambion | AM9932 |

saliva using a pre-calibrated plastic bulb into a vial containing 1200 ul of DRUL buffer (Table 1). Samples were stored and transported at room temperature.

## SARS-CoV-2 assay

In early experiments, RNA was extracted using a modified phenol-chloroform extraction method. 80 μl of 3M sodium acetate, pH 5.5 was added to 800 μl of sample plus buffer and mixed. Then 800 μl of acid-phenol:chloroform pH 4.5 (with IAA, 125:24:1, Ambion, Cat# 9720) was added and mixed. Samples were centrifuged at 12,000 x g for 10 minutes at 4°C after which the aqueous phase (750 μl) was placed into a new tube. 750 μl of OmiPur chloroform: Iso-Amyl Alcohol (Calbiochem, Cat# 3155) was added, mixed, then centrifuged at 12,000 x g for 10 minutes at 4°C. The aqueous phase (550 μl) was placed into a new tube to which 2 μl GlycoBlue (Invitrogen, Cat# AM9515) was added and mixed. 550 μl of ice cold 100% isopropanol was then added and incubated for 15 minutes at -80°C or overnight at -20°C. Samples were centrifuged at 20,000 x g for 20 minutes at 4°C and supernatant removed without disturbing the pellet. 1 ml of cold 75% ethanol was added to the pellet and centrifuged at 20,000 x g for 5 minutes at 4°C. The supernatant was removed and the pellet dried at room temperature for 10 minutes and resuspended in 35 μl of nuclease-free water. In later experiments, RNA was extracted using a column extraction method with a commercial kit (Qiagen, QIAamp DSP Viral RNA Mini Kit, Cat# 61904) according to the manufacturer's instructions. RNA was eluted in 35 μl of nuclease-free water.

cDNA was amplified using TaqPath 1 Step RT-PCR (Life Tech, Cat# A15300) with CDC validated primers and probes (IDT, CDC Emergency Use Authorization Kit) using the Bio-Rad CFX96 C1000 Touch Real-Time PCR Detection System. Samples were considered interpretable if the housekeeping control (RNase P) cycle threshold (Ct) was less than 40 and viral RNA was considered detected with both viral primers/probes (N1 and N2) at Ct <40.

To determine the limit of detection (LOD) of the DRUL saliva assay, contrived clinical specimens (found to be viral-free in the absence of synthetic RNA) were made by spiking in known amounts of quantitative synthetic RNA from SARS-Related Coronavirus 2 (BEI Resources, Cat# NR-52358) into 300 μl of saliva added to indicated amounts of DRUL buffer. Saliva collected from normal volunteers previously determined to be negative for SARS-CoV-2 was pooled and spiked with DRUL buffer containing synthetic SARS-CoV-2 RNA (BEI Resources, Cat # 52358).

To assess sensitivity and specificity of the DRUL saliva assay, thirty NP swab samples were obtained from New York City Public Health Laboratory (NYC PHL). The NP swabs were collected using standard methods by a provider and placed in 3 ml of VTM, and 300 μl of the VTM was added to 1200 μl DRUL buffer at NYC PHL and then transported to the Darnell laboratory for testing.

To determine the ability of DRUL buffer to inactivate virus, Huh-7.5 cells were plated at $1.67 \times 10^5$ per well in each well of 6 well plates and allowed to adhere overnight. Human coronavirus 229E ($3.66 \times 10^6$ PFU/ml) was used as a surrogate for SARS-CoV-2. Mixtures of DRUL

buffer and virus at volume ratios of 1:2, 1:3, 1:4, 1:5, 1:6 and 1:10 were incubated overnight and added to the Huh-7.5 cells the following morning. The viability of the Huh-7.5 cells was assessed after 3 and 5 days of incubation, yielding approximate $TCID_{50}$ values. The $TCID_{50}$ was calculated as the concentration of virus that when diluted in a defined concentration of DRUL buffer led to 50% viability of Huh-7.5 cells on day 3 post inoculation.

To measure viral RNA stability in DRUL buffer, specified concentrations of human coronavirus 229E were incubated with saliva and DRUL buffer and assayed for presence of viral RNA after overnight incubation or after seven days at room temperature and at 0˚C, 25˚C, and 38˚C for seven days. cDNA was amplified using iScript Reverse Transcription Supermix (BioRad, Cat#1708841) and two primer sets for human coronavirus 229E, set 1: forward-`TGAAGATGC` `TTGTACTGTGGCT` and reverse-`CTGTCATGTTGCTCATGGGG`, set 2 forward-`AGATGCTTGT` `ACTGTGGCTTCT` and reverse-`GTCATGTTGCTCATGGGGGAG` (IDT, custom) from 5' to 3' [8] and FASTSTART Universal SYBR Green Master Mix (Millipore Sigma, Cat # 4913914001) using the Bio-Rad CFX96 C1000 Touch Real-Time PCR Detection System. Samples were considered interpretable if the house keeping control (beta-actin) Ct was <40 and viral RNA was considered detected if Ct for both viral primers were <40.

## Statistical analysis

Mean cycle thresholds and standard deviations were calculated in determining and confirming the LOD and in describing the stability of RNA in buffer. Pearson's correlation was used to describe the correlation between NP swab samples assayed with the Cepheid Xpert Xpress SARS-CoV-2 platform and with the DRUL assay.

## Results

### DRUL saliva assay validation

To establish the LOD of the DRUL saliva assay, simulated specimen matrix was made using 5 pooled saliva samples and DRUL buffer spiked with 10-fold serial dilutions of synthetic SARS-CoV-2 RNA. Samples were extracted using a phenol-chloroform or column-based method. Dilutions were tested in triplicate at each concentration of viral RNA. The LOD was determined to be 1 copy/μl with both extraction methods (Fig 1A and 1B). The LOD was confirmed with 20 replicates, each spiked with 2, 1, and 0.2 copies/μl of synthetic RNA, using both extraction methods (Fig 1C and 1D) and with 20 separate extractions, each spiked with 2, 1, and 0.2 copies/μl of synthetic RNA using the column-based extraction method (Fig 1E).

Given a paucity of positive samples to use for a clinical evaluation study, we created simulated positive samples representing various viral concentrations. Synthetic viral RNA was spiked into five individual specimen matrices at 2, 4, 6, 8, 10, and 100 times the confirmed LOD (1 copy/μl) to simulate a range of viral load. RNA was extracted using the phenol or column-based method. In addition, 10 negative specimen matrices were assayed. As expected, mean Cts decreased with increasing viral RNA concentrations using both extraction methods (Fig 2A and 2B). RNase P Cts was <40 and Cts were similar in all samples.

To compare the DRUL saliva assay with a clinically validated platform, we obtained 30 NP swab samples that had tested positive with a wide range of Cts (17.3 to 39.5 on the N2 target) on the Cepheid Xpert Xpress SARS-CoV-2 assay [11]. We detected 30/30 positives (100% sensitivity), and comparison of the Ct values of the N2 target on both platforms revealed that they were highly correlated (Fig 3, Pearson correlation, $R^2$ = 0.96). These results indicate that the DRUL saliva assay performed with high specificity, and with quantitative results that were concordant over a wide range (4.8 x $10^6$-fold) of detectable viral RNA in clinical samples, as confirmed by comparison to the Cepheid Xpert Xpress SARS-CoV-2 assay.

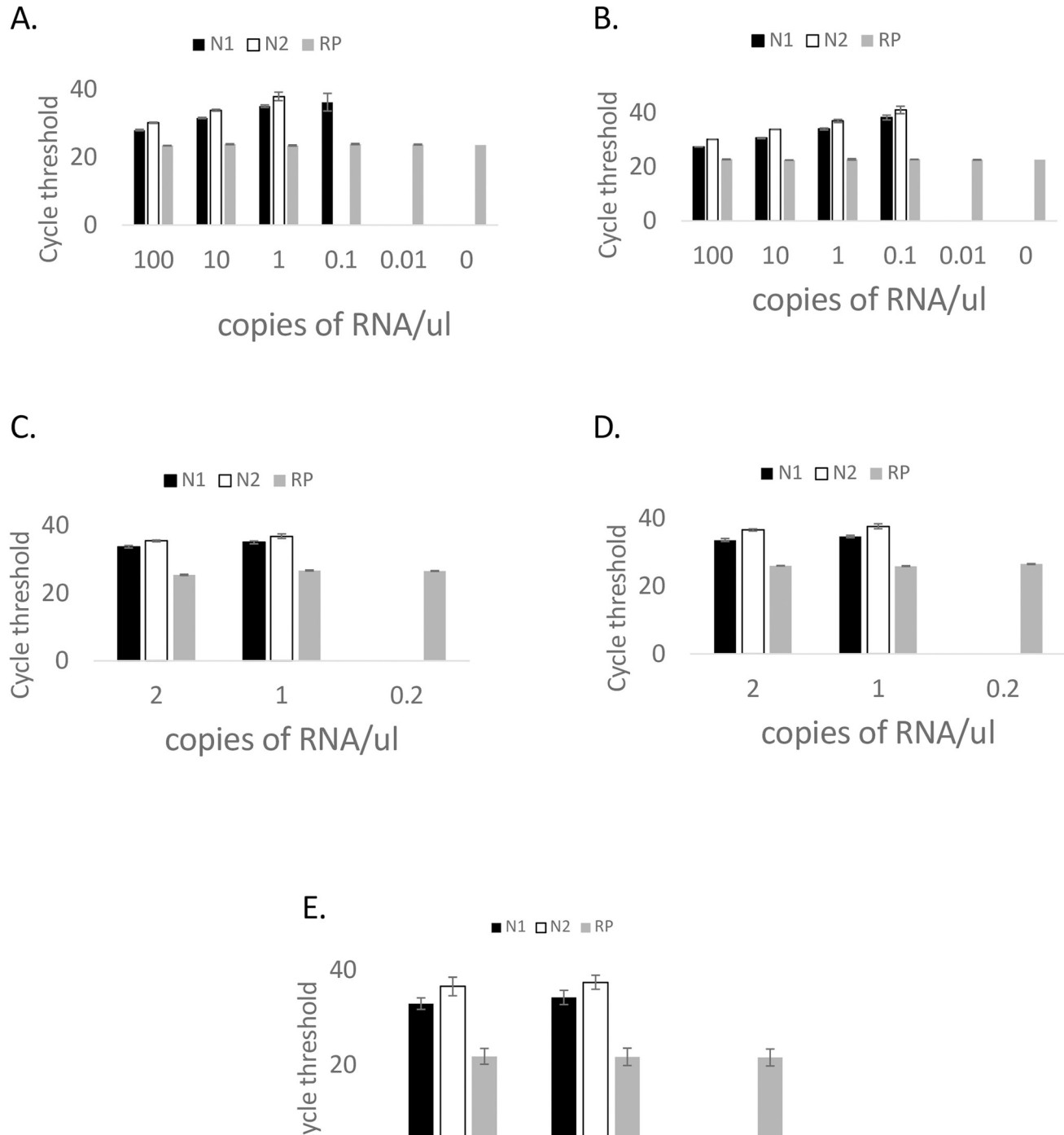

**Fig 1. Limit of detection (LOD).** Determination of LOD with (A) phenol and (B) column-based extraction methods and confirmation of LOD using (C) phenol and (D) column-based extraction methods in 20 replicates and (E) column-based extraction method in 20 separate extractions from unique saliva samples. Black bars = N1 primer, open bars = N2 primer, gray bars = RNase P primer. Error bars = 1 standard deviation.

A

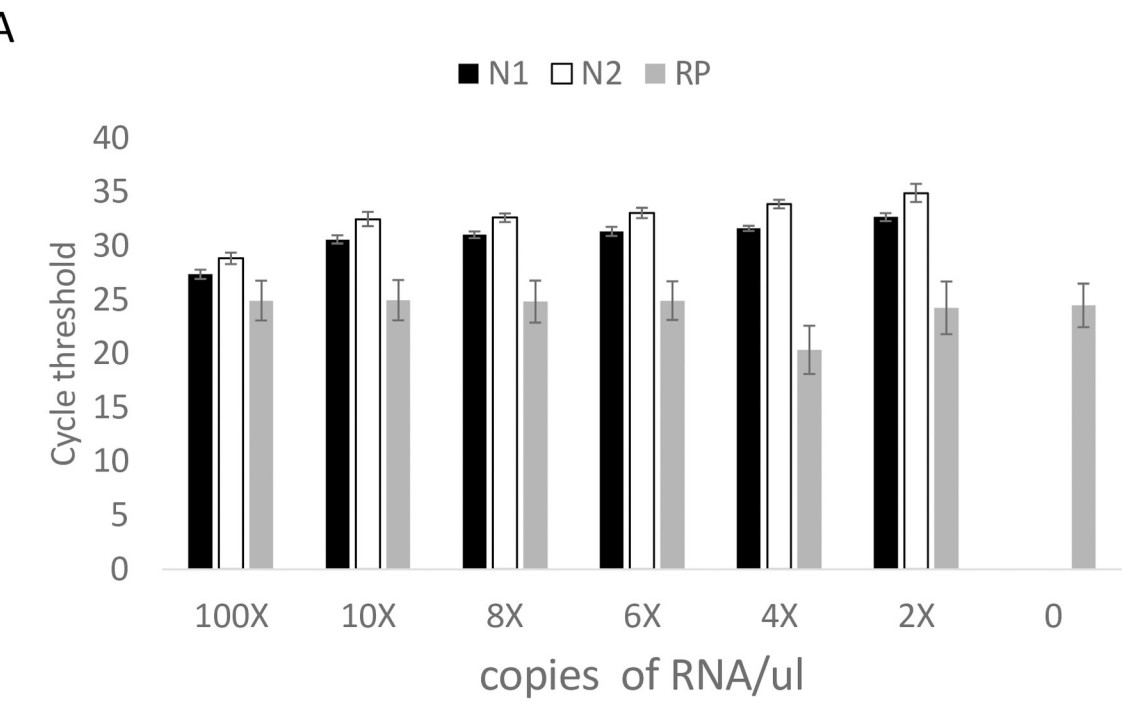

B

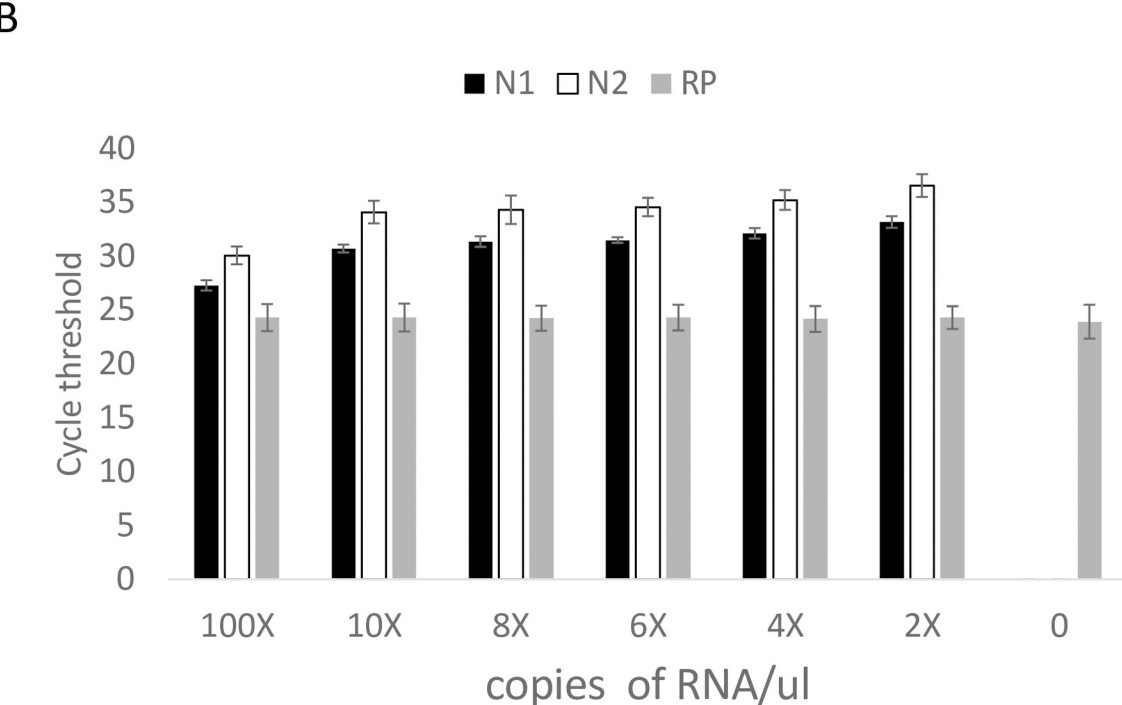

**Fig 2. Assay performance with two extraction methods.** Specimen matrix spiked with specified concentration of synthetic RNA and extracted using A. phenol and B. column-based methods. Black bars = N1 primer, open bars = N2 primer, gray bars = RNase P primer. Error bars = 1 standard deviation.

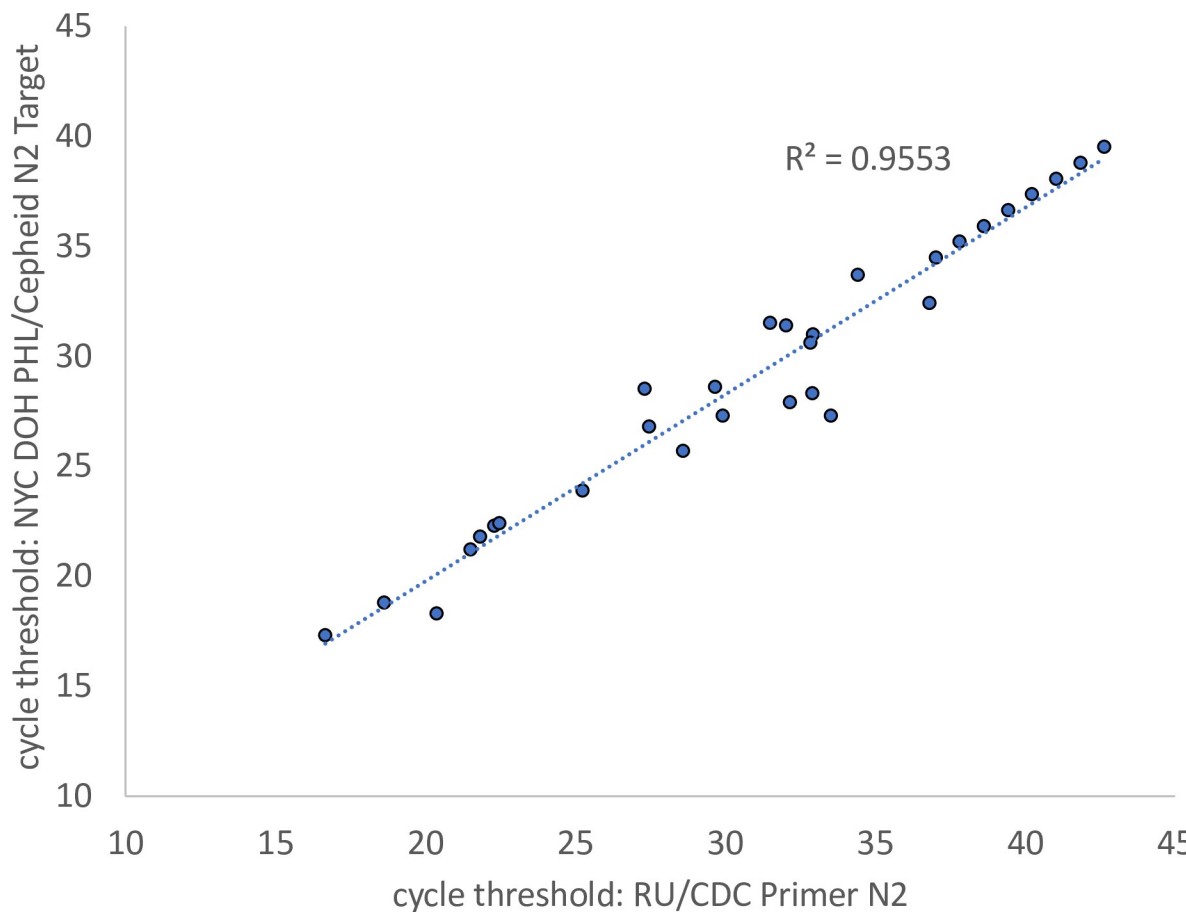

**Fig 3. Correlation of NP samples on Cepheid Xpert Xpress SARS-Cov-2 platform versus phenol extraction in DRUL buffer.**

To further investigate the sensitivity and specificity of the DRUL saliva assay, we compared 63 OP swab results collected from RU Occupational Health Services, which were tested at commercial laboratory and with co-collected self-collected saliva samples in DRUL buffer. The majority of co-collected specimens (57/63, 90.5%) were negative by both assays (Table 2). Of the remaining six specimens, SARS-CoV-2 RNA was detected in the saliva specimen in three participants. Of these three, two of the co-collected OP specimens were negative and one was indeterminate by the Commercial Laboratory A test. These individuals were symptomatic. Three additional participants were negative by OP swab, and the saliva test was invalid. RNaseP target was not detected, which we most commonly found correlated with insufficient saliva specimen, although in some cases inhibitors may have been present.

In a second study to assess sensitivity and specificity, we compared 99 NP swab samples collected by healthcare providers at New York City Health and Hospital–Elmhurst (tested at

**Table 2. Comparison of SARS-CoV-2 DRUL assay with commercially tested OP swabs.**

|  |  | Darnell Lab-Saliva | | |
|---|---|---|---|---|
|  |  | **Positive** | **Negative** | **Invalid** |
| Commercial Lab A- | Positive | 0 | 0 | 0 |
| OP swab | Negative | 2 | 57 | 3 |
|  | Inconclusive | 1 | 0 | 0 |

**Table 3. Comparison of SARS-CoV-2 DRUL assay with commercially tested NP swabs.**

| | | Darnell Lab-Saliva | |
|---|---|---|---|
| | | Positive | Negative |
| Commercial Lab B- | Positive | 0 | 0 |
| NP swab | Negative | 1 | 98 |

Commercial Laboratory B) with co-collected, self-collected saliva samples in DRUL buffer (Table 3). All samples but one were negative by both assays. The DRUL saliva assay identified one positive sample of the 99 which was negative by NP swab in an asymptomatic individual. At the time these experiments were done, the turnaround time for results from paired samples in Commercial Laboratory B was three to five days, while results from the DRUL saliva assay were generally available the next day, including the one positive sample. These studies taken together suggest similar, if not higher sensitivity of the DRUL saliva assay than commonly accepted viral assays using OP or NP swabs.

To assess the stability of viral RNA in DRUL buffer, we titrated concentrations of human coronavirus 229E into saliva and DRUL buffer and compared Ct values of samples incubated overnight or after 7 days at room temperature. There was no significant difference between the Ct values of samples incubated overnight and those incubated for 7 days (Fig 4A and 4B). We further assessed the stability of viral RNA incubated at 0°C, 25°C, and 38°C for 7 days, to mimic potential temperature ranges during sample transport and did not find a significant difference in Ct values (Fig 4C).

To evaluate the effect of DRUL buffer on viral infectivity, we used human coronavirus 229E as a surrogate for SARS-CoV-2. We assessed the viability of Huh-7.5 cells, a well characterized, adult hepatocellular carcinoma cell line, after exposure to various dilutions of coronavirus (stock $3.66 \times 10^6$ PFU/ml) in DRUL buffer. Huh-7.5 cell survival indicates that the virus was inactivated by the DRUL buffer, and cells remained viable after exposure of stock virus diluted with DRUL buffer at ratios of 1:4 (DRUL:virus), indicating that DRUL buffer completely inactivates virus at $2.75 \times 10^6$ PFU/ml (Fig 5A and 5B). At DRUL to virus ratio of 1:5 ($2.93 \times 10^6$ PFU/ml), approximately half the Huh-7.5 cells were lysed at day 5 indicating viral survival, and this value was taken as a conservative estimate of the $TCID_{50}$. We compared this with the AVL buffer, a component of the QIAamp Viral RNA kit that the CDC determined to inactivate virus [7]. AVL buffer inactivated virus at a buffer to virus ratio of 1:3 ($2.44 \times 10^6$ PFU/ml) but not at 1:4, indicating that DRUL buffer inactivates virus at a level comparable to AVL buffer (Fig 5A and 5B). In control samples, cell lysis occurred when we exposed Huh-7.5 cells to as little as 4 PFU/ml human coronavirus E229 without DRUL buffer, and no cell lysis occurred when DRUL buffer was added without virus.

To determine the minimum incubation time required for the DRUL buffer to inactivate virus, we incubated DRUL buffer and virus at a ratio of 1:4 for 60 minutes, 10 minutes and 10 seconds before incubating with Huh-7.5 cells. We found that 100% of the Huh-7.5 cells were viable at 3 and 5 days after incubation with virus exposed to DRUL buffer for as little as 10 seconds (Fig 4C). Taken together these results indicate that DRUL buffer nearly instantly inactivates live, high titer coronavirus.

## Clinical use of the DRUL saliva assay

These validation data were submitted to New York State CLEP and the DRUL saliva assay was subsequently approved for use as a clinical diagnostic test. The assay was used in 3,724 samples between May and October of 2020 from individuals who ranged in age from 3 months to 92

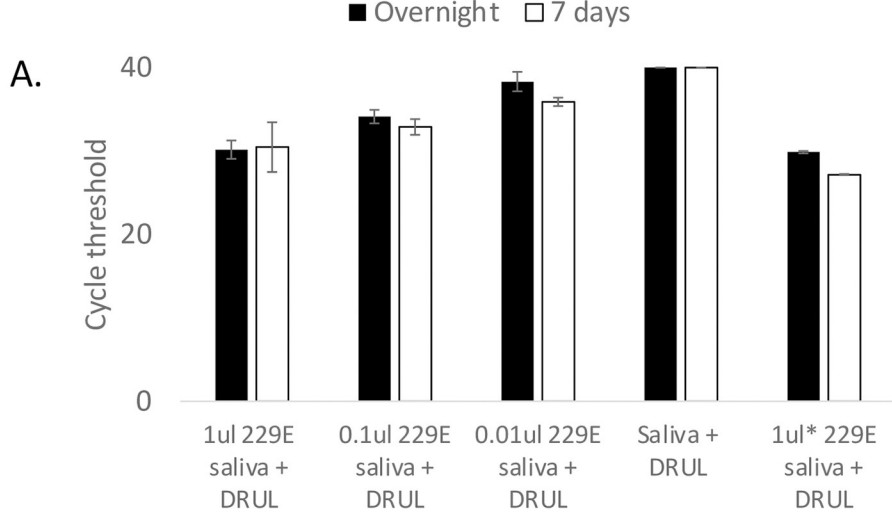

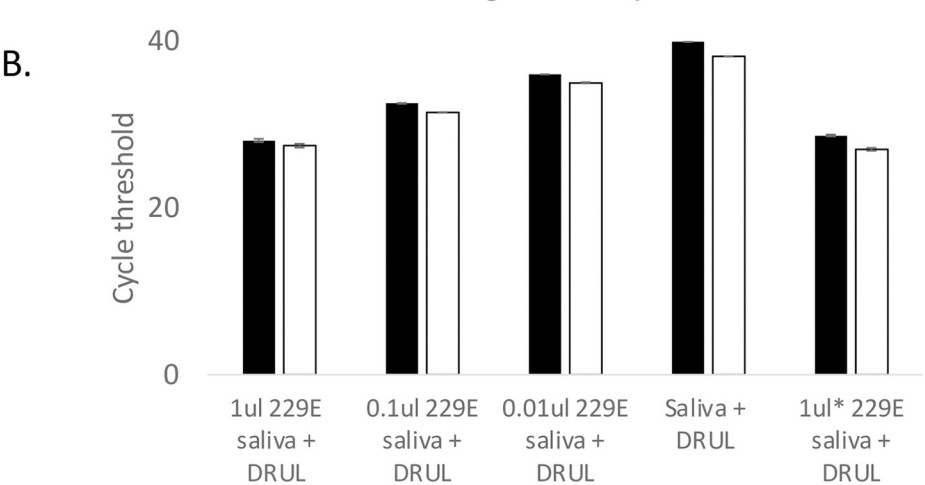

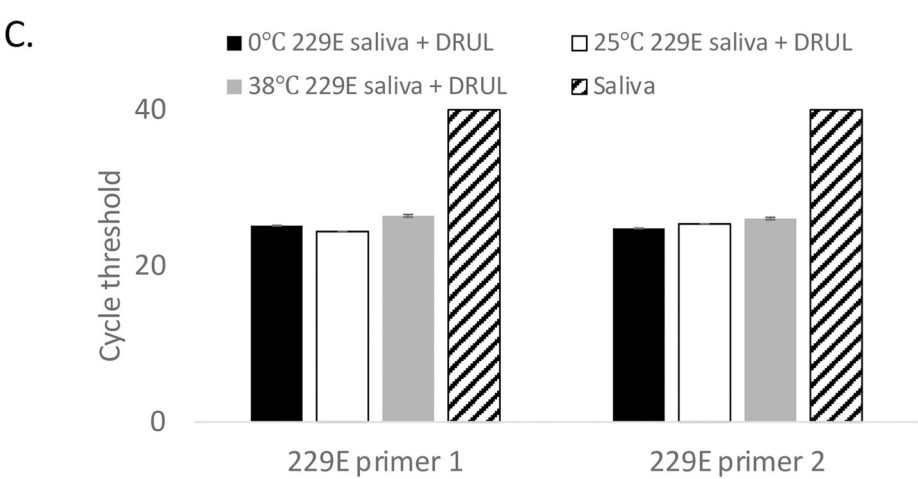

**Fig 4. Stability of viral RNA.** Stability of RNA assessed with (A) primer set 1 and (B) primer set 2 in DRUL buffer. Black bars = overnight incubation, open bars = after 7 days of incubation. C. Stability assessed at 0˚C (black bars), 25˚C (open bars), and 38˚C (gray bars) and saliva alone (striped bars). Error bars = 1 standard deviation.

years. We began with testing symptomatic employees and asymptomatic essential employees coming onto the RU campus.

In July of 2020, the RU Child and Family Center (CFC) for children of employees between the ages of three months and five years reopened on a pilot basis, enrolling 58 children in July and August of 2020, then 87 children starting in September. Each child, teacher and staff member was tested weekly, and parents were also offered testing. 2117 kits were distributed over 12 weeks (S1 Table), which were typically taken home, where saliva was collected and added to DRUL buffer with a plastic bulb syringe. Electronic sample submission forms linked to a personalized registration data were completed for each sample, and tubes returned to RU the following day.

Over these 12 weeks, only one asymptomatic parent tested positive. The parent was isolated, the child (a contact) was quarantined, and the classroom closed. Overall, 26 children school days were missed (number of children in room x number of days classroom closed or school days missed; Table 4). All other tests among the children, teachers, staff, and parents were negative, allowing these rooms to remain open, consistent with (or more conservative than) NYS/NYC DOE school guidance. There were three additional room closures due to symptomatic (as defined by CDC guidelines) children or teachers who tested negative, resulting in 46 children school days missed. There were 72 missed children school days out of 4205 (1.7%) over the course of 12 weeks.

## Discussion

Here, we report the validation of the DRUL saliva assay for SARS CoV-2 molecular testing as performed at RU. This assay was easy to administer, using a self-collection kit that could be performed at home by adults or by older children under adult supervision. RT-PCR assays, using either traditional phenol-chloroform or column-based extraction methods revealed that the assay was extremely sensitive, with a LOD of 1 copy/μl of viral RNA, and was found to perform nearly identically to a clinical platform (Cepheid Xpert Xpress SARS-CoV-2 assay). Moreover, the assay was found to be at least as sensitive as OP and NP swabs assessed by commercial laboratories using FDA approved molecular tests.

The DRUL saliva assay was developed with the goal of overcoming early obstacles to widespread SARS-CoV-2 testing, such as shortages of reagents and specialized supplies, healthcare provider access, and PPE for healthcare providers. This method also limits potential exposure during transit and of laboratory personnel during performance of the assay. The $TCID_{50}$ of virus (~2.64 x$10^6$ PFU/ml) diluted 1:4 (v:v) in DRUL buffer was found to compare favorably to commercial inactivation buffers (Qiagen's AVL buffer, Fig 5 or the SDNA-1000 saliva collection device with an estimated $TCID_{50}$ of ~1 x $10^4$ PFU/ml used by Rutgers Clinical Genomics Laboratory TaqPath SARS-CoV-2 Assay) [10, 12].

Such solutions have a health hazard label (2) that grades them as less toxic than household bleach. DRUL buffer kits are distributed with appropriate cautions and instructions on what to do in case of a spill or contact. To further minimize risk, we have recently succeeded in decreasing the required volume of DRUL from 1200 μl to 300 μl with similar results (unpublished data). To date there have been no adverse events reported from the use of the DRUL saliva assay.

A.

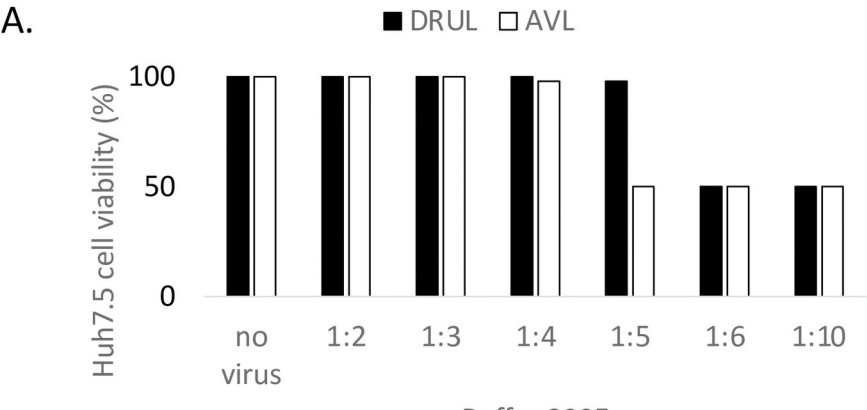

| Dilution Factor (Buffer:Virus) | Conc. assayed (x10^6 PFU/ml) |
|---|---|
| 2 | 1.83 |
| 3 | 2.44 |
| **4** | **2.75** |
| 5 | 2.93 |
| 6 | 3.05 |
| 10 | 3.29 |

B.

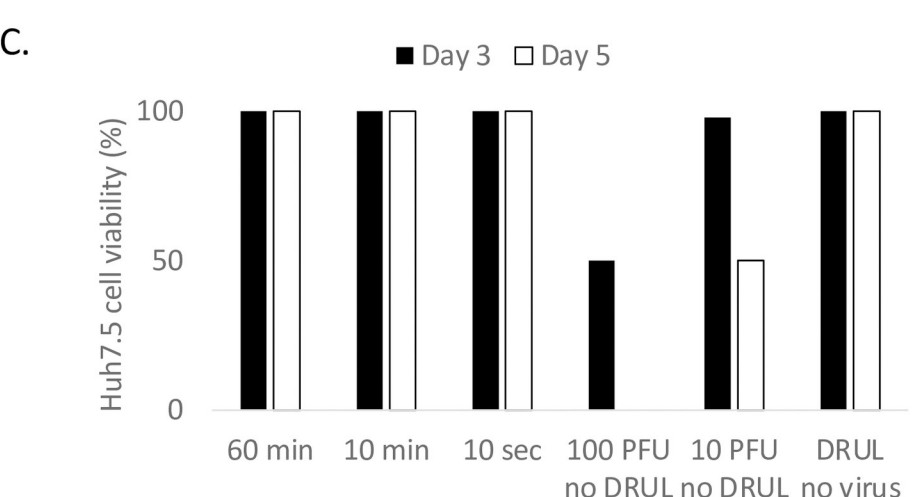

C.

**Fig 5. Inactivation of virus in DRUL buffer.** Huh-7.5 cell lysis assessed at day 3 (A) and day 5 (B) after incubation with DRUL (black bars) or AVL buffer (open bars) and human coronavirus E229 at buffer:virus volume ratios ranging from 1:2 to 1:10. Insert table shows viral concentration at each ratio. C. Huh-7.5 cell lysis assessed at days 3 (black bars) and 5 (open bars) after incubation with human coronavirus E229 exposed to DRUL buffer for 60 minutes, 10 minutes, or 10 seconds.

**Table 4. Child and family center closures.**

| Closure event | Cause | Total days closed (no.) | School days closed (no.) | Children school days missed (no.) |
|---|---|---|---|---|
| 1 | symptomatic child | 6 | 4 | 28 |
| 2 | positive parent | 6 | 4 | 26* |
| 3 | symptomatic teacher | 3 | 1 | 10 |
| 4 | symptomatic child | 4 | 2 | 8 |

*includes 14-day quarantine by child of the positive parent

The DRUL saliva assay was used for testing symptomatic individuals and screening asymptomatic essential employees on the RU campus over the course of 6 months. It proved easy to use across a variety of ages and individuals with varied backgrounds. The CDC Guidance considers viral testing strategies in partnership with schools at the K-12 level as part of a comprehensive COVID-19 prevention approach for safely keeping schools open [13]. They discuss the utility of testing in both diagnostic testing for close contacts or symptomatic students, teachers, and staff and screening among teachers and staff at all community transmission levels and students as well at moderate, substantial, and high transmission levels. In addition, testing is recommended for participation in low, intermediate, or high-risk sports at any community transmission risk level.

Although testing is not currently included in the CDC Guidance for child care centers [14], our experience with the assay proved of value to aid in the reopening of not only of the RU childcare center but here and in subsequent work (unpublished data) in reopening of the entire RU campus community. The use of the test minimized the number of days a classroom closed and allowed the rest of the center to remain open safely. With testing, 98.7% student attendance was possible, along with reassurance that both they and their teachers had undetectable viral RNA on a weekly basis. The interpretation of the test results in the context of the clinical history of symptoms, exposure, and travel was key in the effective use of the test results. We are currently validating and implementing a sample pooling strategy to increase testing capacity as well as decrease testing resources as a potentially efficient testing strategies in these settings. As SARS-CoV-2 infection, particularly in the unvaccinated child population, remains a significant clinical issue, the DRUL saliva test offers a simple, safe, and cost-effective method for use as part of highly scalable "back to work/school" strategies.

## Supporting information

**S1 Table. Kits distributed to children, teachers, staff, and parents at the Child and Family Center.**
(TIF)

**S1 Fig. Instructions for self-saliva collection included with the kit.**
(TIF)

**S1 Data.**
(XLSX)

## Acknowledgments

We would like to thank all volunteers for their participation in our research study.

## Author Contributions

**Conceptualization:** Mayu O. Frank, Robert B. Darnell.

**Data curation:** Mayu O. Frank, Nathalie E. Blachere, Salina Parveen, John Fak, Ann Campbell, Marissa Bergh.

**Formal analysis:** Mayu O. Frank, Nathalie E. Blachere, Ezgi Hacisuleyman, Randal C. Fowler, Robert B. Darnell.

**Investigation:** Mayu O. Frank, Nathalie E. Blachere, Salina Parveen, Ezgi Hacisuleyman, John Fak, Joseph M. Luna, Eleftherios Michailidis, Samara Wright, Pamela Stark, Ann Campbell, Ashley Foo, Thomas P. Sakmar, Virginia Huffman, Marissa Bergh, Audrey Goldfarb, Andres Mansisidor, Agata L. Patriotis, Karl H. Palmquist, Nicolas Poulton, Rachel Leicher, César D. M. Vargas, Irene Duba, Arlene Hurley, Joseph Colagreco, Nicole Pagane, Dana E. Orange, Kevin Mora, Jennifer L. Rakeman, Randal C. Fowler, Helen Fernandes, Michelle F. Lamendola-Essel, Nicholas Didkovsky, Leopolda Silvera, Joseph Masci, Machelle Allen, Charles M. Rice, Robert B. Darnell.

**Methodology:** Nathalie E. Blachere, Salina Parveen, Ezgi Hacisuleyman, John Fak, Joseph M. Luna, Eleftherios Michailidis, Jennifer L. Rakeman, Michelle F. Lamendola-Essel, Charles M. Rice, Robert B. Darnell.

**Project administration:** Mayu O. Frank.

**Resources:** Robert B. Darnell.

**Software:** Nicole Pagane, Nicholas Didkovsky.

**Supervision:** Helen Fernandes.

**Validation:** Mayu O. Frank, Nathalie E. Blachere, Salina Parveen, Helen Fernandes.

**Writing – original draft:** Mayu O. Frank, Robert B. Darnell.

**Writing – review & editing:** Mayu O. Frank, Nathalie E. Blachere, Salina Parveen, Ezgi Hacisuleyman, Joseph M. Luna, Eleftherios Michailidis, Samara Wright, Pamela Stark, Ann Campbell, Ashley Foo, Thomas P. Sakmar, Virginia Huffman, Marissa Bergh, Audrey Goldfarb, Andres Mansisidor, Agata L. Patriotis, Karl H. Palmquist, Nicolas Poulton, Rachel Leicher, César D. M. Vargas, Irene Duba, Arlene Hurley, Joseph Colagreco, Nicole Pagane, Dana E. Orange, Kevin Mora, Jennifer L. Rakeman, Randal C. Fowler, Helen Fernandes, Michelle F. Lamendola-Essel, Nicholas Didkovsky, Leopolda Silvera, Joseph Masci, Machelle Allen, Charles M. Rice, Robert B. Darnell.

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
