## [Decision Letter · Decision Letter 0]

11 May 2021

PONE-D-21-11188

DRUL for School:  Opening Pre-K with safe, simple, sensitive saliva testing for SARS-CoV-2

PLOS ONE

Dear Dr. Darnell,

Thank you for submitting your manuscript to PLOS ONE. After careful consideration, we feel that it has merit but does not fully meet PLOS ONE’s publication criteria as it currently stands. Therefore, we invite you to submit a revised version of the manuscript that addresses the points raised during the review process.

We look forward to receiving your revised manuscript.

Kind regards,

Kanhaiya Singh, Ph.D

Academic Editor

PLOS ONE

Additional Editor Comments:

Although the article was well received by the reviewers, they have suggested some changes to make this study robust.

Journal Requirements:

"C.M.R acknowledges the generous support of the G. Harold and Leila Y. Mathers Charitable Foundation, the BAWD Foundation, and The Rockefeller University. R.B.D. wishes to disclose that he receives consulting fees as a Senior Visiting Fellow at MITRE Corporation, has started a charitable LLC, D4S Testing, to offer free DRUL saliva testing to NYC school children, and is an Investigator of the Howard Hughes Medical Institute."

3. We noted in your submission details that a portion of your manuscript may have been presented or published elsewhere. [No, but submitted to MedRxiv] Please clarify whether this [conference proceeding or publication] was peer-reviewed and formally published. If this work was previously peer-reviewed and published, in the cover letter please provide the reason that this work does not constitute dual publication and should be included in the current manuscript.

4. Please include captions for *all* your Supporting Information files at the end of your manuscript, and update any in-text citations to match accordingly. Please see our Supporting Information guidelines for more information: http://journals.plos.org/plosone/s/supporting-information.

Reviewers' comments:

Reviewer's Responses to Questions

**Comments to the Author**

1. Is the manuscript technically sound, and do the data support the conclusions?

Reviewer #1: Yes

Reviewer #2: Yes

Reviewer #3: Yes

2. Has the statistical analysis been performed appropriately and rigorously? 

Reviewer #1: Yes

Reviewer #2: Yes

Reviewer #3: No

3. Have the authors made all data underlying the findings in their manuscript fully available?

Reviewer #1: No

Reviewer #2: Yes

Reviewer #3: No

4. Is the manuscript presented in an intelligible fashion and written in standard English?

Reviewer #1: Yes

Reviewer #2: Yes

Reviewer #3: Yes

5. Review Comments to the Author

Reviewer #1: I read with great interest the article from Darnell lab titled, 'DRUL for School: Opening Pre-K with safe, simple, sensitive saliva testing for SARS-CoV-2' It provides a practical solution to the current problem with opening of preK schools in the post-pandemic era.

The authors have shown scientific rigor and sound statistical analysis in the current study. It is written in a lucid manner.

Here are a few suggestions I would like to make to the authors:

1. References: i) ref 8,9,10 are missing in the introduction

ii) Sequence of references should also be looked into.

2. CDC has updated guidelines / operational strategies for K-12 schools to reopen. https://www.cdc.gov/coronavirus/2019-ncov/community/schools-childcare/operation-strategy.html?CDC_AA_refVal=https%3A%2F%2Fwww.cdc.gov%2Fcoronavirus%2F2019-ncov%2Fcommunity%2Fschools-childcare%2Fschools.html

A pertinent question to answer for the current manuscript is 'How does the DRUL saliva method fit in the context of these current guidelines?' especially considering the low risk of transmission amongst pre-K population.

3. The following points may be added to the discussion to make the utility DRUL saliva test more relevant in context of opening of schools:

i) Screening testing is particularly valuable in areas with moderate, substantial, and high levels of community transmission. Screening testing for K–12 schools may allow schools to move between different testing strategies as community prevalence (and therefore risk assessment) changes.

ii) Pooled tests for a cohort (pre-K students) may prove to be a feasible strategy. This approach increases the number of individuals that can be tested and reduces the need for testing resources.

Reviewer #2: It was of an immense pleasure for me to go through this article of Darnell Lab. It indeed addresses the need for simple, safe, sensitive, and scalable SARS-CoV-2 tests in this on going pandemic. I find this article to have a good statistical base and is scientifically developed.

Here are a few suggestions which I would like to bring into consideration:

1. TITLE: DRUL for School: Opening Pre-K with safe, simple, sensitive saliva testing for SARS-COV2.

even though this assay was used to aid in the reopening of a childcare center that enrolled children as young as three months old, it was used across a variety of ages and individuals with varied backgrounds; this title somewhat downplays its wide utility in various other workplaces where it can be of an exceptional use.

2. The DRUL buffer is based on the solution widely used in RNA extraction that contains [12].

This sentence on page no. 2 seems incomplete and needs appropriate completion.

3. Also references 9, 10 cannot be found mentioned in the content of this manuscript.

4. Antigen-based testing but not real-time polymerase chain reaction correlates with severe acute respiratory syndrome coronavirus 2 viral culture. Pekosz et al. Clinical Infectious Diseases (January 20, 2021).

This aspect can be further taken into consideration with a goal of increasing the PPV of the screening tests.

5. Also, instead of administering this test as a form of screening to each child, teacher and staff member weekly, along with inclusion of parents, it will be worthwhile to further assess the advantages of this method when the same assay is offered to test the pooled samples of small cohorts in a large group, or when clustered sampling can be done, in the setting of low prevalence of this infection.

Reviewer #3: The paper by Frank et al. aims at developing a PCR test using a self collected saliva sample kit at home for the detection of SARS-CoV-2.

Major Points:1, Please provide the figure2 to figure5 with the manuscript, as they are missing.

2, Please mention which statistical tools were used to analyze the data.

Minor Points: How long the individuals are instructed to avoid eating or using cleansing agents before collecting the samples? 30 mints ( mentioned in Specimen collection : Methods section) or 1 hour ( Mentioned in supplementary figure 1)

6. PLOS authors have the option to publish the peer review history of their article (what does this mean?). If published, this will include your full peer review and any attached files.

Reviewer #1: No

Reviewer #2: No

Reviewer #3: No

---

## [Author Response · Author response to Decision Letter 0]

17 May 2021

Reviewer #1: 

I read with great interest the article from Darnell lab titled, 'DRUL for School: Opening Pre-K with safe, simple, sensitive saliva testing for SARS-CoV-2' It provides a practical solution to the current problem with opening of preK schools in the post-pandemic era.

The authors have shown scientific rigor and sound statistical analysis in the current study. It is written in a lucid manner.

Here are a few suggestions I would like to make to the authors:

1. References: i) ref 8,9,10 are missing in the introduction

ii) Sequence of references should also be looked into.

Numbering of the references have been reviewed and corrected.

2. CDC has updated guidelines / operational strategies for K-12 schools to reopen. https://www.cdc.gov/coronavirus/2019-ncov/community/schools-childcare/operation-strategy.html?CDC_AA_refVal=https%3A%2F%2Fwww.cdc.gov%2Fcoronavirus%2F2019-ncov%2Fcommunity%2Fschools-childcare%2Fschools.html

A pertinent question to answer for the current manuscript is 'How does the DRUL saliva method fit in the context of these current guidelines?' especially considering the low risk of transmission amongst pre-K population.

Thank you for this comment. At the link above, the CDC recognizes the importance of viral testing strategies in partnership with schools at the K-12 level as a comprehensive COVID-19 prevention approach to safely keeping schools open. They discuss the utility of testing in both diagnostic testing for close contacts or symptomatic students, teachers, and staff and screening among teachers and staff at all community transmission levels and students as well at moderate, substantial, and high transmission levels. In addition, testing is recommended for participation in low, intermediate, or high-risk sports at any community transmission risk level. This is now added to the discussion. The CDC guidance in child care programs does not include viral testing although it acknowledges that children in child care setting can become infected and spread COVID-19 to others (https://www.cdc.gov/coronavirus/2019-ncov/community/schools-childcare/guidance-for-childcare.html). In our experience, screening has been an important tool to identify asymptomatic members of the school community (e.g. a parent of a child at the child care center as presented) as positive, allowing the school to close the classroom quickly while conducting additional testing. Classrooms have then re-opened when it was determined that the child of the parent who was positive was a contact and not positive him/herself. 

3. The following points may be added to the discussion to make the utility DRUL saliva test more relevant in context of opening of schools:

i) Screening testing is particularly valuable in areas with moderate, substantial, and high levels of community transmission. Screening testing for K–12 schools may allow schools to move between different testing strategies as community prevalence (and therefore risk assessment) changes.

This has been added to the discussion section.

ii) Pooled tests for a cohort (pre-K students) may prove to be a feasible strategy. This approach increases the number of individuals that can be tested and reduces the need for testing resources.

Thank you for this comment. We are currently working on validating and implementing a pooling strategy to improve testing efficiency and this has been added to the discussion section.

Reviewer #2: 

It was of an immense pleasure for me to go through this article of Darnell Lab. It indeed addresses the need for simple, safe, sensitive, and scalable SARS-CoV-2 tests in this on going pandemic. I find this article to have a good statistical base and is scientifically developed.

Here are a few suggestions which I would like to bring into consideration:

1. TITLE: DRUL for School: Opening Pre-K with safe, simple, sensitive saliva testing for SARS-COV2.

even though this assay was used to aid in the reopening of a childcare center that enrolled children as young as three months old, it was used across a variety of ages and individuals with varied backgrounds; this title somewhat downplays its wide utility in various other workplaces where it can be of an exceptional use.

We very much appreciate the Reviewer’s supportive comment. We have modified the text to address this point (below), while also maintaining our focus on PreK. Our rationale for the latter is that young children are likely to be the last to be vaccinated, and our work from this study and subsequent findings in the RU PreK cohort demonstrating asymptomatic infection and transmission in a preK 7 month old (Singer et al, Lancet Infectious Disease, in review) emphasize the critical nature of PreK asymptomatic testing.

We have added the following text to the Discussion:

“Although testing is not currently included in the Guidance for child care centers [14], our experience with the assay proved of value to aid in the reopening of not only of the RU childcare center but here and in subsequent work (unpublished data) in reopening of the entire RU campus community.” 

2. The DRUL buffer is based on the solution widely used in RNA extraction that contains [12].

This sentence on page no. 2 seems incomplete and needs appropriate completion.

The buffer contains guanidinium thiocyanate. This is corrected in the manuscript.

3. Also references 9, 10 cannot be found mentioned in the content of this manuscript.

This has been corrected.

4. Antigen-based testing but not real-time polymerase chain reaction correlates with severe acute respiratory syndrome coronavirus 2 viral culture. Pekosz et al. Clinical Infectious Diseases (January 20, 2021).

This aspect can be further taken into consideration with a goal of increasing the PPV of the screening tests.

Thank you for this comment. We appreciate this point, but believe the issues raised here and in the broader literature are significantly more complex than the Pekosz et al paper alone can address, and believe a more detailed discussion, while warranted, is beyond the scope of this clinical report.

In the Pekosz article, the authors uses positive viral cultures after exposure to positive specimens as a surrogate for infectiousness and transmissibility and find that positive antigen test results more closely align with positive cultures as compared to PCR test results (PPV 90.0 vs 73.7% for antigen vs PCR testing as compared to positive cultures). They note that the PCR positive samples that did not infect the cultured cells were obtained from patients whose samples were obtained a week after symptom onset and the estimated viral load was significantly lower in these samples, indicating that these patients continue to have a positive PCR test even when they are no longer infectious, potentially leading to unnecessary isolation. 

The presence of viral RNA that can be detected by PCR even after one is no longer infectious is an important issue, one that the CDC addresses by recommending against retesting those who may have been re-exposed to COVID-19 within 90 days of symptom onset of the initial infection.1 If a PCR test is done post infection, clearly, clinical history the including symptom onset is critical to properly interpret the result. 

The highly sensitive nature of the PCR test allows also means however, that it may be possible to detect virus in people with potentially lower viral loads such as when they are pre-symptomatic. They may not be detected by antigen testing due to the test’s lower sensitivity (41.2%) as compared to PCR testing among asymptomatic people.2 The Reviewer may also wish to review the work of Joe DeRisi and colleagues from HHMI,3 who note that “Direct antigen assays, such as Binax-CoV2, are unlikely to rival the sensitivity of RT-PCR.”, and moreover found that only 1/11 samples positive by PCR at Ct > 29 could be detected by antigen testing. In settings such as work and schools, a highly sensitive test such as a PCR test may be more useful in detecting positives among asymptomatic individuals, as we have found in our own unpublished studies documenting asymptomatic carrier and transmission status from a 7 month old in preK to adults who then got ill (e.g. Singer et al., Lancet Inf Disease, in review). Moreover, we have identified positives among those who were asymptomatic at testing but later became symptomatic. Our ability to detect these individuals leads to faster isolation and less exposure. The article by Pekosz et al. makes a good point that clinical history is critical in interpreting the results of the PCR test and this has been added to the discussion. 

1. https://www.cdc.gov/coronavirus/2019-ncov/hcp/duration-isolation.html

2. https://www.cdc.gov/mmwr/volumes/69/wr/mm695152a3.htm#T2_down

3. https://pubmed.ncbi.nlm.nih.gov/33173911/

5. Also, instead of administering this test as a form of screening to each child, teacher and staff member weekly, along with inclusion of parents, it will be worthwhile to further assess the advantages of this method when the same assay is offered to test the pooled samples of small cohorts in a large group, or when clustered sampling can be done, in the setting of low prevalence of this infection.

This appreciate this point and are currently working extending the work reported here toward implementing a pooling strategy. This has been added to the discussion. 

Reviewer #3:

The paper by Frank et al. aims at developing a PCR test using a self collected saliva sample kit at home for the detection of SARS-CoV-2.

Major Points:

1, Please provide the figure2 to figure5 with the manuscript, as they are missing.

 Figures are now submitted.

2, Please mention which statistical tools were used to analyze the data.

 A statistical analysis section is added.

Minor Points: How long the individuals are instructed to avoid eating or using cleansing agents before collecting the samples? 30 mints ( mentioned in Specimen collection : Methods section) or 1 hour ( Mentioned in supplementary figure 1)

Individuals were asked to wait 30 minutes. Supplementary Figure 1 is corrected.

---

## [Decision Letter · Decision Letter 1]

26 May 2021

DRUL for School:  Opening Pre-K with safe, simple, sensitive saliva testing for SARS-CoV-2

PONE-D-21-11188R1

Dear Dr. Darnell,

We’re pleased to inform you that your manuscript has been judged scientifically suitable for publication and will be formally accepted for publication once it meets all outstanding technical requirements.

Kind regards,

Kanhaiya Singh, Ph.D

Academic Editor

PLOS ONE

Additional Editor Comments (optional):

Please address the minor comments made by Reviewer 3 during the proofreading stage.

Reviewers' comments:

Reviewer's Responses to Questions

**Comments to the Author**

1. If the authors have adequately addressed your comments raised in a previous round of review and you feel that this manuscript is now acceptable for publication, you may indicate that here to bypass the “Comments to the Author” section, enter your conflict of interest statement in the “Confidential to Editor” section, and submit your "Accept" recommendation.

Reviewer #1: All comments have been addressed

Reviewer #2: All comments have been addressed

Reviewer #3: All comments have been addressed

2. Is the manuscript technically sound, and do the data support the conclusions?

Reviewer #1: Yes

Reviewer #2: Yes

Reviewer #3: Yes

3. Has the statistical analysis been performed appropriately and rigorously? 

Reviewer #1: Yes

Reviewer #2: Yes

Reviewer #3: Yes

4. Have the authors made all data underlying the findings in their manuscript fully available?

Reviewer #1: Yes

Reviewer #2: Yes

Reviewer #3: Yes

5. Is the manuscript presented in an intelligible fashion and written in standard English?

Reviewer #1: Yes

Reviewer #2: Yes

Reviewer #3: Yes

6. Review Comments to the Author

Reviewer #1: All concerns raised by the previous reviewers have been addressed very nicely.

To be honest, reviewing this paper has been a privileged learning experience for me.

Such data needs to come out in public domain at earliest.

Thank you.

Reviewer #2: (No Response)

Reviewer #3: Author addressed all the comments. Author should correct the minor issues in the manuscript before submitting the final version to the journal

e.g:

1,At DRUL buffer to virus ratio 1:5 concentration, 3.41 x 106 PFU/ml viral particles mentioned in Results section, but table in Figure 5 predicts different number.

2, Reference 14 is missing.

7. PLOS authors have the option to publish the peer review history of their article (what does this mean?). If published, this will include your full peer review and any attached files.

Reviewer #1: No

Reviewer #2: No

Reviewer #3: No

---

## [Editor Report · Acceptance letter]

18 Jun 2021

PONE-D-21-11188R1 

DRUL for School:  Opening Pre-K with safe, simple, sensitive saliva testing for SARS-CoV-2 

Dear Dr. Darnell:

I'm pleased to inform you that your manuscript has been deemed suitable for publication in PLOS ONE. Congratulations! Your manuscript is now with our production department. 

Kind regards, 

on behalf of

Dr. Kanhaiya Singh 

Academic Editor

PLOS ONE